# Generalized Clustering by Learning to Optimize Expected Normalized Cuts

## Abstract

We introduce a novel end-to-end approach for learning to cluster in the absence of labeled examples. Our clustering objective is based on optimizing normalized cuts, a criterion which measures both intra-cluster similarity as well as inter-cluster dissimilarity. We define a differentiable loss function equivalent to the expected normalized cuts. Unlike much of the work in unsupervised deep learning, our trained model directly outputs final cluster assignments, rather than embeddings that need further processing to be usable. Our approach generalizes to unseen datasets across a wide variety of domains, including text, and image. Specifically, we achieve state-of-the-art results on popular unsupervised clustering benchmarks (e.g., MNIST, Reuters, CIFAR-10, and CIFAR-100), outperforming the strongest baselines by up to 10.9%. Our generalization results are superior (by up to 21.9%) to the recent top-performing clustering approach with the ability to generalize.

## 1 Introduction

Clustering unlabeled data is an important problem from both a scientific and practical perspective. As technology plays a larger role in daily life, the volume of available data has exploded. However, labeling this data remains very costly and often requires domain expertise. Therefore, unsupervised clustering methods are one of the few viable approaches to gain insight into the structure of these massive unlabeled datasets.

One of the most popular clustering methods is spectral clustering (Shi & Malik, 2000; Ng et al., 2002; Von Luxburg, 2007), which first embeds the similarity of each pair of data points in the Laplacian's eigenspace and then uses k-means to generate clusters from it. Spectral clustering not only outperforms commonly used clustering methods, such as k-means (Von Luxburg, 2007), but also allows us to directly minimize the pairwise distance between data points and solve for the optimal node embeddings analytically. Moreover, it is shown that the eigenvector of the normalized Laplacian matrix can be used to find the approximate solution to the well known normalized cuts problem (Ng et al., 2002; Von Luxburg, 2007).

In this work, we introduce *CNC*, a framework for *Clustering* by learning to optimize expected *Normalized Cuts*. We show that by directly minimizing a continuous relaxation of the normalized cuts problem, *CNC* enables end-to-end learning approach that outperforms top-performing clustering approaches. We demonstrate that our approach indeed can produce lower normalized cut values than the baseline methods such as SpectralNet, which consequently results in better clustering accuracy.

Let us motivate *CNC* through a simple example. In Figure 1, we want to cluster 6 images from CIFAR-10 dataset into two clusters. The affinity graph for these data points is shown in Figure 1(a) (details of constructing such graph is discussed in Section 4.2). In this example, it is obvious that the optimal clustering is the result of cutting the edge connecting the two triangles. Cutting this edge will result in the optimal value for the normalized cuts objective. In *CNC*, we define a new differentiable loss function equivalent to the expected normalized cuts objective. We train a deep learning model to minimize the proposed loss in an unsupervised manner without the need for any labeled datasets. Our trained model directly returns the probabilities of belonging to each cluster (Figure 1(b)). In this example, the optimal normalized cuts is 0.286 (Equation 1), and as we can see, the *CNC* loss also converges to this value (Figure 1(c)).

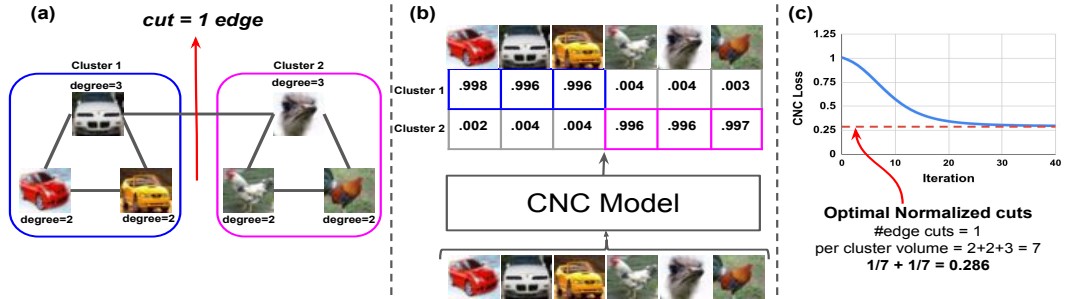

Figure 1: Motivational example: (a) affinity graph of 6 images from CIFAR-10, the objective is to cluster these images into two clusters. (b) *CNC* model is trained to minimize expected normalized cuts in an unsupervised manner without the need for any labeled data. For each data point, our model directly outputs the probabilities of it belonging to each of the clusters. (c) The *CNC* loss converges to the optimal normalized cuts value. In Algorithm 1 we show how we can scale this approach through a batch processing technique to large datasets.

We compare the performance of *CNC* to several learning-based clustering approaches (Spectral-Net (Shaham et al., 2018), DEC (Xie et al., 2016), DCN (Yang et al., 2017), VaDE (Jiang et al., 2017), DEPICT (Ghasedi Dizaji et al., 2017), IMSAT (Hu et al., 2017), and IIC (Ji et al., 2019)) on four datasets: MNIST, Reuters, CIFAR10, and CIFAR100. Our results show up to 10.9% improvement over the baselines. Moreover, generalizing spectral embeddings to unseen data points, a task commonly referred to as out-of-sample-extension (OOSE), is a non-trivial task (Bengio et al., 2003; Belkin et al., 2006; Mendoza Quispe et al., 2016). Our results confirm that *CNC* generalizes to unseen data. Our generalization results are superior (by up to 21.9%) to SpectralNet (Shaham et al., 2018), the recent top-performing clustering approach with the ability to generalize.

## 2 RELATED WORK

Recent deep learning approaches to clustering attempt to embed the input data into a form that is amenable to clustering by k-means or Gaussian Mixture Models. (Yang et al., 2017; Xie et al., 2016) focused on learning representations for clustering. To find the clustering-friendly latent representations and to better cluster the data, DCN (Yang et al., 2017) proposed a joint dimensionality reduction (DR) and K-means clustering approach in which DR is accomplished via learning a deep neural network. DEC (Xie et al., 2016) simultaneously learns cluster assignment and the underlying feature representation by iteratively updating a target distribution to sharpen cluster associations.

Several other approaches rely on a variational autoencoder that utilizes a Gaussian mixture prior (Jiang et al., 2017; Dilokthanakul et al., 2016; Hu et al., 2017; Ji et al., 2019; Ben-Yosef & Weinshall, 2018). These approaches are mainly based on data augmentation, where the network is trained to maximize the mutual information between inputs and predicted clusters, while regularizing the network so that the cluster assignment of the data points is consistent with the assignment of the augmented points.

Different clustering objectives, such as self-balanced k-means and balanced min-cut, have also been exhaustively studied (Liu et al., 2017; Chen et al., 2017; Chang et al., 2014). One of the most effective techniques is spectral clustering, which first generates node embeddings in the eigenspace of the graph Laplacian, and then applies k-means clustering to these vectors (Shi & Malik, 2000; Ng et al., 2002; Von Luxburg, 2007). To address the fact that clusters with the lowest graph conductance tend to have few nodes (Leskovec, 2009; Zhang & Rohe, 2018), (Zhang & Rohe, 2018) proposed regularized spectral clustering to encourage more balanced clusters.

Generalizing clustering to unseen nodes and graphs is nontrivial (Bengio et al., 2003; Belkin et al., 2006; Mendoza Quispe et al., 2016). A recent work, SpectralNet (Shaham et al., 2018), takes a deep learning approach to spectral clustering that generalizes to unseen data points. This approach first learns embeddings of the similarity of each pair of data points in Laplacian's eigenspace and then applies k-means to those embeddings to generate clusters. Unlike SpectralNet, we propose an end-to-end learning approach with a differentiable loss that directly minimizes the normalized cuts. We show that our approach indeed can produce lower normalized cut values than the baseline methods such as SpectralNet, which consequently results in better clustering accuracy. Our evaluation results show that *CNC* improves generalization accuracy on unseen data points by up to 21.9%.

## 3 PRELIMINARIES

Since *CNC* objective is based on optimizing normalized cuts, in this section, we briefly overview the formal definition of this metric.

### 3.1 FORMAL DEFINITION OF NORMALIZED CUTS

Let $G = (V, E, W)$ be a graph where $V = \{v_i\}$ and $E = \{e(v_i, v_j)|v_i \in V, v_j \in V\}$ are the set of nodes and edges in the graph and $w_{ij} \in W$ is the edge weight of the $e(v_i, v_j)$. Let $n$ be the number of nodes. A graph $G$ can be clustered into $g$ disjoint sets $S_1, S_2, \ldots S_g$, where the union of the nodes in those sets are $V$ ($\bigcup_{k=1}^{g} S_k = V$), and each node belongs to only one set ($\bigcap_{k=1}^{g} S_k = \emptyset$), by simply removing edges connecting those sets. For example, in Figure 1(a), by removing one edge two disjoint clusters are formed.

Normalized cuts (*Ncuts*) which is defined based on the graph conductance, has been studied by (Shi & Malik, 2000; Zhang & Rohe, 2018), and the cost of a cut that forms disjoint sets $S_1, S_2, \ldots S_g$ is computed as:

$$Ncuts(S_1, S_2, \ldots S_g) = \sum_{k=1}^{g} \frac{cut(S_k, \bar{S}_k)}{vol(S_k, V)} \tag{1}$$

Where $\bar{S}_k$ represents the complement of $S_k$, i.e., $\bar{S}_k = \bigcup_{i \neq k} S_i$. $cut(S_k, \bar{S}_k)$ is called *cut* and is the total weight of the edges that are removed from $G$ in order to form disjoint sets $S_k$ and $\bar{S}_k$. $vol(S_k, V)$ is the total edge weights ($w_{ij}$), whose end points ($v_i$, or $v_j$) belong to $S_k$. The *cut* and *vol* are:

$$cut(S_k, \bar{S}_k) = \sum_{v_i \in S_k, v_j \in \bar{S}_k} w_{ij} \quad , \quad vol(S_k, V) = \sum_{v_i \in S_k} \sum_{v_j \in V} w_{ij} \tag{2}$$

Note that in Equation 2, $S_k$ and $\bar{S}_k$ are disjoint, i.e., $S_k \cap \bar{S}_k = \emptyset$, while in *vol*, $S_k \subset V$. In running example (Figure 1), since the edge weights are one, $cut(S_1, \bar{S}_1) = cut(S_2, \bar{S}_2) = 1$, and $vol(S_1, V) = vol(S_2, V) = 2 + 2 + 3 = 7$. Thus the $Ncuts(S_1, S_2) = \frac{1}{7} + \frac{1}{7} = 0.286$. In this example one can see that such clustering results in minimum value of the normalized cuts. *CNC* aims to find a cut that the normalized cuts (Equation 1) is minimized.

## 4 CNC FRAMEWORK

Finding the cluster assignments that minimizes the normalized cuts is NP-complete and an approximation to the this problem is based on the eigenvectors of the normalized graph Laplacian which has been studied in (Shi & Malik, 2000; Zhang & Rohe, 2018). *CNC*, on the other hand, is a neural network framework for learning to cluster in the absence of labeled examples by directly minimizing the continuous relaxation of the normalized cuts. As shown in Algorithm 1, end-to-end training of the *CNC* contains two steps, i.e, (i) data points embedding (line 3), and (ii) clustering (lines 4-9). In data points embedding, the goal is to learn embeddings that capture the affinity of the data points, while the clustering step uses those embeddings to learn the CNC model and outputs the cluster assignments. Next, we first focus on the clustering step and we introduce our new differentiable loss function to train CNC model. Later in Section 4.2, we discuss the details of the embedding step.

### 4.1 CLUSTERING STEP: LEARN CNC MODEL

In this section, we describe the clustering step in Algorithm 1 (lines 4-9). For each data point $x_i$, the input to clustering step is embedding $v_i \in \mathbb{R}^d$ (detail in Section 4.2). The goal is to learn CNC model $F_\theta : \mathbb{R}^d \to \mathbb{R}^g$ that for a given embedding $v_i \in \mathbb{R}^d$ it returns $y_i = F_\theta(v_i) \in \mathbb{R}^g$, which represents the assignment probabilities over $g$ clusters. Clearly for $n$ data points, it returns $Y \in \mathbb{R}^{n \times g}$ where $Y_{ik}$ represents the probability that $v_i$ belongs to cluster $S_k$. The CNC model $F_\theta$ is implemented using a neural network, where the parameter vector $\theta$ denotes the network weights. We propose a loss function based on output $Y$ to calculate the expected normalized cuts. Thus *CNC* learns the $F_\theta$ by minimizing this loss (Equation 7).

Recall that $cut(S_k, \bar{S}_k)$ is the total weight of the edges that are removed from $G$ in order to form disjoint sets $S_k$ and $\bar{S}_k$. In our setup, embeddings are the nodes in graph $G$, and neighbors of an embedding $v_i$ are based on the k-nearest neighbors. Let $Y_{ik}$ be the probability that node $v_i$ belongs to cluster $S_k$. The probability that node $v_j$ does not belong to $S_k$ would be $1 - Y_{jk}$. Therefore, $\mathbb{E}[cut(S_k, \bar{S}_k)]$ can be formulated by Equation 3, where $\mathcal{N}(v_i)$ is the set of nodes adjacent to $v_i$.

---

**Algorithm 1**
End-to-End Training of *CNC*: *Clustering* by learning to optimize expected *Normalized Cuts*

---

1: **Input:** dataset $X \subseteq \mathbb{R}^m$, number of clusters $g$, data point embedding size $d$, batch size $b$
2: **Output:** Cluster assignments of data points.
   ***Preprocessing step, learn data points embedding (details in Section 4.2):***
3: Given a dataset $X = \{x_1, \ldots x_n\}$, train a Siamese network to find embeddings $\{v_1, \ldots v_n\}$, $v_i \in \mathbb{R}^d$ that represent the affinity of the data points. $G_{\theta_{\text{siamese}}} : \mathbb{R}^m \to \mathbb{R}^d$
   ***Clustering step, learn CNC model $F_\theta$ (details in Section 4.1):***
4: **while** CNC loss in Equation 6 not converged **do**
5:    Sample a random minibatch $M$ of size $b$ from the embeddings
6:    Compute affinity graph $W \in \mathbb{R}^{b \times b}$ over the $M$ based on the k-nearest neighbors
7:    Use $M$ and $W$ to train CNC model $F_\theta : \mathbb{R}^d \to \mathbb{R}^g$ that minimizes the expected normalized cuts (Equation 6) via backpropagation. For a data point with embedding $v_i$ the output $y_i = F_\theta(v_i)$ represents the assignment probabilities over $g$ clusters.
8: **end while**
   ***Inference, cluster assignments***
9: For every data points $x_i$ whose embedding is $v_i$ return $\arg\max$ of $y_i = F_\theta(v_i)$ as its cluster assignment.

---

$$\mathbb{E}_{Y \sim F_\theta}[cut(S_k, \bar{S}_k)] = \sum_{v_i \in S_k} \sum_{v_j \in \mathcal{N}(v_i)} w_{ij} Y_{ik}(1 - Y_{jk}) \tag{3}$$

Since the weight matrix $W$ represents the edge weights adjacent nodes, we can rewrite Equation 3:

$$\mathbb{E}_{Y \sim F_\theta}[cut(S_k, \bar{S}_k)] = \underbrace{\sum}_{\text{reduce-sum}} Y_{:,k}(1 - Y_{:,k})^\intercal \odot W \tag{4}$$

The element-wise product with the weight matrix $(\odot W)$ ensures that only the adjacent nodes are considered. Moreover, the result of $Y_{:,k}(1 - Y_{:,k})^\intercal \odot W$ is an $n \times n$ matrix and *reduce-sum* is the sum over all of its elements. From Equation 2, $vol(S_k, V)$ is the total edge weights ($w_{ij}$), whose end points ($v_i$, or $v_j$) belong to $S_k$. Let $D$ be a column vector of size $n$ where $D_i$ is the total edge weights from node $v_i$. We can update Equation 3 as follows to find the expected normalized cuts.

$$\mathbb{E}_{Y \sim F_\theta}[Ncuts] = \sum_{k=1}^{g} \sum_{v_i \in S_k} \sum_{v_j \in \mathcal{N}(v_i)} \frac{w_{ij} Y_{ik}(1 - Y_{jk})}{\sum_{v_l \in V} Y_{lk} D_l} \tag{5}$$

The matrix representation is given in Equation 6, where $\Gamma = Y^\intercal D$ is a vector in $\mathbb{R}^g$, and $g$ is the number of sets/clusters. $\oslash$ is element-wise division and the result of $(Y \oslash \Gamma)(1 - Y)^\intercal \odot W$ is a $n \times n$ matrix where *reduce-sum* is the sum over all of its elements.

$$
\begin{aligned}
\mathbb{E}_{Y \sim F_\theta}[Ncuts] &= \underbrace{\sum}_{\text{reduce-sum}} \sum_{k=1}^{g} \frac{Y_{:,k}}{\Gamma_k}(1 - Y_{:,k})^\intercal \odot W \\
&= \underbrace{\sum}_{\text{reduce-sum}} (Y \oslash \Gamma)(1 - Y)^\intercal \odot W
\end{aligned}
\tag{6}
$$

CNC model $F_\theta$ is implemented using a neural network, where the parameter $\theta$ denotes the network weights ($y_i = F_\theta(v_i)$). *CNC* is trained to optimize Equation 7 via backpropagation (Algorithm 1).

$$\arg\min_\theta \underbrace{\sum}_{\text{reduce-sum}} (Y \oslash \Gamma)(1 - Y)^\intercal \odot W \tag{7}$$

As you can see the affinity graph $W$ is part of the *CNC* loss (Equation 7). Clearly, when the number of data points ($n$) is large, such calculation can be expensive. However, in our experimental results, we show that for large dataset (e.g., Reuters contains 685,071 documents), it is possible to optimize the loss on randomly sampled minibatches of data. We also build the affinity graph over a given minibach using the embeddings and based on their k nearest-neighbor (Algorithm 1 (lines 5-6)). Specifically, in our implementation, *CNC* model $F_\theta$ is a fully connected layer followed by gumble softmax, trained on randomly sampled minibatches of data to minimize Equation 6. In Section 5.7 through a sensitivity analysis we show that the minibatch size affects the accuracy of our model. When training is over, the final assignment of a data point with embedding $v_i$ to a cluster is the $\arg\max$ of $y_i = F_\theta(v_i)$ (Algorithm 1 (line 9)).

## 4.2 EMBEDDING STEP

In this section, we discuss the embedding step (line 3 in Algorithm 1). Different affinity measures, such as simple euclidean distance or nearest neighbor pairs combined with a Gaussian kernel, have been used in spectral clustering. Recently it is shown that unsupervised application of a Siamese network to determine the distances improves the quality of the clustering (Shaham et al., 2018).

In this work, we also use Siamese networks to learn embeddings that capture the affinities of the data points. Siamese network is trained to learn an adaptive nearest neighbor metric. It learns the affinities directly from euclidean proximity by "labeling" points $x_i$, $x_j$ positive if $\|x_i - x_j\|$ is small and negative otherwise. In other words, it generates embeddings such that adjacent nodes are closer in the embedding space and non-adjacent nodes are further. Such network is typically trained to minimize contrastive loss:

$$L_{\text{siamese}} = \begin{cases} ||v_i - v_j||^2, & (x_i, x_j) \text{ is a positive pair} \\ \max(1 - ||v_i - v_j||^2, 0)^2, & (x_i, x_j) \text{ is a negative pair} \end{cases}$$

where $v_i = G_{\theta_{\text{siamese}}}(x_i)$, and $G_{\theta_{\text{siamese}}} : \mathbb{R}^m \to \mathbb{R}^d$ is a Siamese network that transforms representations in the input space $x_i \in \mathbb{R}^m$ to embeddings $v_i \in \mathbb{R}^d$.

## 5 EXPERIMENTS

The main goals of our experiments are to evaluate: (a) The performance of *CNC* against the existing clustering approaches. (b) The ability of *CNC* to generalize to unseen data compared to the top-performing generalizable baseline. (c) The effectiveness of minimizing Normalized cuts on improving the clustering results. (d) The generalization performance of *CNC* as we vary the number of data points in training dataset.

### 5.1 DATASETS AND BASELINE METHODS

We evaluate the performance of *CNC* in comparison to several deep learning-based clustering approaches on four real world datasets: MNIST, Reuters, CIFAR-10, and CIFAR-100. The details of the datasets are as follows:

- MNIST is a collection of 70,000 28×28 gray-scale images of handwritten digits, divided into 60,000 training images and 10,000 test images.

- The Reuters dataset is a collection of English news labeled by category. Like SpectralNet, DEC, and VaDE, we used the following categories: corporate/industrial, government/social, markets, and economics as labels and discarded all documents with multiple labels. Each article is represented by a tfidf vector using the 2000 most frequent words. The dataset contains 685,071 documents. We divided the data randomly to a 90%-10% split to evaluate the generalization ability of *CNC*. We also investigate the imapact of training data size on the generalization by considering following splits: 90%-10%, 70%-30%, 50%-50%, 20%-80%, and 10%-90%.

- CIFAR-10 consists of 60000 32x32 colour images in 10 classes, with 6000 images per class. There are 50000 training images and 10000 test images.

- CIFAR-100 has 100 classes containing 600 images each with a 500/100 train/test split per class.

In all runs we assume the number of clusters is given. In MNIST and CIFAR-10 number of clusters (g) is 10, g = 4 in Reuters, g = 100 in CIFAR-100. We compare *CNC* to SpectralNet (Shaham et al., 2018), DEC (Xie et al., 2016), DCN (Yang et al., 2017), VaDE (Jiang et al., 2017), DEPICT (Ghasedi Dizaji et al., 2017), IMSAT (Hu et al., 2017), and IIC (Ji et al., 2019). While (Yang et al., 2017; Xie et al., 2016) focused on learning representations for clustering, other approaches (Jiang et al., 2017; Dilokthanakul et al., 2016; Hu et al., 2017; Ji et al., 2019; Ben-Yosef & Weinshall, 2018) rely on a variational autoencoder that utilizes a Gaussian mixture prior. SpectralNet (Shaham et al., 2018), takes a deep learning approach to spectral clustering that generalizes to unseen data points. Table 1 shows the results reported for these six methods.

Similar to (Shaham et al., 2018), for MNIST and Reuters we use publicly available and pre-trained autoencoders[1]. The autoencoder used to map the Reuters data to code space was trained based on a random subset of 10,000 samples from the full dataset. Similar to (Hu et al., 2017), for CIFAR-10 and CIFAR-100 we applied 50-layer pre-trained deep residual networks trained on ImageNet to extract features and used them for clustering.

[1] https://github.com/slim1017/VaDE/tree/master/pretrain_weights

| Method | MNIST | | Reuters | | CIFAR-10 | | CIFAR-100 | |
|---|---|---|---|---|---|---|---|---|
| | ACC | NMI | ACC | NMI | ACC | NMI | ACC | NMI |
| DEC | 0.843* | 0.800* | 0.756* | - | 0.469 | - | - | - |
| DCN | 0.830** | 0.810** | - | - | - | - | - | - |
| VaDE | 0.945† | - | 0.794† | - | - | - | - | - |
| DEPICT | 0.965†† | 0.917†† | - | - | - | - | - | - |
| IMSAT | 0.984‡‡ | - | 0.719‡‡ | - | 0.456‡‡ | - | 0.275‡‡ | - |
| IIC | **0.993**††† | - | - | - | 0.617††† | - | - | - |
| SpectralNet | 0.971‡ | **0.924**‡ | 0.803‡ | 0.532‡ | 0.501 | 0.463 | 0.236 | 0.231 |
| *CNC* | 0.972 | **0.924** | **0.824** | **0.583** | **0.702** | **0.586** | **0.345** | **0.502** |

Table 1: Performance of various clustering methods on MNIST, Reuters, CIFAR-10 and CIFAR-100. The model is trained on all data. (–) means values are not reported. (∗) reported in DEC [(Xie et al., 2016)], (∗∗) reported in DCN [(Yang et al., 2017)], (†) reported in VaDE [(Jiang et al., 2017)], (††) reported in DEPICT [(Ghasedi Dizaji et al., 2017)], (‡‡) reported in IMSAT [(Hu et al., 2017)], (‡) reported in SpectralNet [(Shaham et al., 2018)], (†††) reported in IIC [(Ji et al., 2019)].

## 5.2 PERFORMANCE MEASURES

We use two commonly used measures, the unsupervised clustering accuracy (ACC), and the normalized mutual information (NMI) in (Cai et al., 2011) to evaluate the accuracy of the clustering. Both ACC and NMI are in [0, 1], with higher values indicating better correspondence the clusters and the true labels. Note that true labels never used neither in training, nor in test.

*Clustering Accuracy (ACC):* For data points $X = \{x_1, \ldots x_n\}$, let $l = (l_1, \ldots l_n)$ and $c = (c_1, \ldots c_n)$ be the true labels and predicted clusters respectively. The ACC is defined as:

$$ACC(l, c) = \frac{1}{n} \max_{\pi \in \prod} \sum_{i=1}^{n} \mathbb{1}\{l_i = \pi(c_i)\}$$

where $\prod$ is the collection of all permutations of $1, \ldots g$. The optimal permutation $\pi$ can be computed using the Kuhn-Munkres algorithm (Munkres, 1957).

*Normalized Mutual Information (NMI):* Let $I(l; c)$ be the mutual information between $l$ and $c$, and $H(.)$ be their entropy. The NMI is:

$$NMI(l, c) = \frac{I(l; c)}{\max\{H(l), H(c)\}}$$

## 5.3 EXPERIMENTAL RESULTS

For each dataset we trained a Siamese network (Hadsell et al., 2006; Shaham & Lederman, 2018) to learn embeddings which represents the affinity of data points by only considering the k-nearest neighbors of each data. In Table 1, we compare clustering performance across four benchmark datasets. Since most of the clustering approaches do not generalize to unseen data points, all data has been used for the training (Later in Section 5.4, to evaluate the generalizability we use 90%-10% split for training and testing).

While the improvement of *CNC* is marginal over MNIST, it performs better across other three datasets. Specifically, over CIFAR-10, *CNC* outperforms SpectralNet and IIC on ACC by 20.1% and 10.9% respectively. Moreover, the NMI is improved by 12.3%. The results over Reuters, and CIFAR-100, show 0.021% and 11% improvement on ACC. The NMI is also 27% better over CIFAR-100. The fact that our *CNC* outperforms existing approaches in most datasets suggests the effectiveness of using our deep learning approach to optimize normalized cuts for clustering.

We performed an ablation study to evaluate the impact of embeddings by omitting this step in Algorithm 1. We find that on both MNIST and Reuters datasets, adding the embedding step improves the performance, but *CNC* without embeddings still outperforms SpectralNet without embeddings. On MNIST, the ACC and NMI are 0.945 and 0.873, whereas with the embeddings, ACC and NMI increase to 0.972 and 0.924 (Table 1). Without embeddings, *CNC* outperforms SpectralNet (with ACC of 0.8 and NMI of 0.814). On Reuters, the ACC and NMI are 0.684 and 0.428, whereas with the embeddings, ACC and NMI increase to 0.824 and 0.583. Again, even without embeddings, *CNC* outperforms SpectralNet (with ACC of 0.605 and NMI of 0.401).

| Method | MNIST | | Reuters | | CIFAR-10 | | CIFAR-100 | |
|---|---|---|---|---|---|---|---|---|
| | ACC | NMI | ACC | NMI | ACC | NMI | ACC | NMI |
| SpectralNet | 0.970[‡] | 0.925[‡] | 0.798[‡] | 0.536[‡] | 0.491 | 0.478 | 0.229 | 0.230 |
| *CNC* | **0.971** | **0.925** | **0.824** | **0.586** | **0.701** | **0.585** | **0.343** | **0.526** |

Table 2: Generalization of clustering methods on MNIST, Reuters, CIFAR-10 and CIFAR-100 datasets. The model is trained only on training set and the reported numbers are the test accuracy. (–) means values are not reported. (‡) reported in SpectralNet [(Shaham et al., 2018)].

| | MNIST | Reuters | CIFAR-10 | CIFAR-100 |
|---|---|---|---|---|
| SpectralNet | 0.913 | 0.351 | 4.229 | 82.831 |
| *CNC* | **0.879** | **0.21** | **2.451** | **58.535** |

Table 3: Numerical value of the normalized cut (Equation 1) over the clustering results of the *CNC* and SpectralNet [(Shaham et al., 2018)]. *CNC* is able to find better cuts than the SpectralNet

## 5.4 GENERALIZATION

We further evaluate the generalization ability of *CNC* by dividing the data randomly to a 90%-10% split and training on the training set and report the ACC and NMI on the test set (Table 2). Among seven methods in Table 1, only SpectralNet is able to generalize to unseen data points. *CNC* outperforms SpectralNet in most datasets by up to 21.9% on ACC and up to 10.7% on NMI. Note that simple arg max over the output of *CNC* retrieves the clustering assignments while SpectralNet relies on k-means to predict the final clusters.

## 5.5 IMPACT OF NORMALIZED CUTS IN CLUSTERING

To evaluate the impact of normalized cuts for the clustering task, we calculate the numerical value of the Normalized cuts (Equation 1) over the clustering results of the *CNC* and SpectralNet. Since such calculation over whole dataset is very expensive we only show this result over the test set.

Table 3 shows the numerical value of the Normalized cuts over the clustering results of the *CNC* and SpectralNet. As one can see *CNC* is able to find better cuts than the SpectralNet. Moreover, we observe that for those datasets that the improvement of the *CNC* is marginal (MNIST and Reuters), the normalized cuts of *CNC* are also only slightly better than the SpectralNet, while for the CIFAR-10 and CIFAR-100 that the accuracy improved significantly the normalized cuts of *CNC* are also much smaller than SpectralNet. The higher accuracy (ACC in Table 2) and smaller normalized cuts (Table 3), verify that indeed *CNC* loss function is a good notion for clustering task.

## 5.6 IMAPACT OF TRAINING DATA SIZE ON THE GENERALIZATION

As you may see in generalization result (Table 2), when we reduce the size of the training data to 90% the accuracy of *CNC* slightly changed in compare to training over the whole data (Table 1). Based on this observation, we next investigate how varying the size of the training dataset affects the generalization. In other words, how ACC and NMI of test data change when we vary the size of the training dataset.

We ran experiment over Routers dataset by dividing the data randomly based on the following data splits: 90%-10%, 70%-30%, 50%-50%, 20%-80%, and 10%-90%. For example, in 10%-90%, we train *CNC* over 10% of the data and we report the ACC and NMI of *CNC* over the 90% test set. Figure 3 shows how the ACC and NMI of *CNC* over the test data change as the size of the training data is varied. For example, when the size of the training data is 90%, the ACC of *CNC* over the test data is 0.824.

As we expected and shown in Figure 3 the ACC and NMI of *CNC* increased as the size of the training data is increased. Interestingly, we observed that with only 10% training data the ACC of *CNC* is 0.68 which is only 14% lower than the ACC with 90% training data. Similarly the NMI of *CNC* with 10% training data is only 18% lower than the NMI with 90% training data.

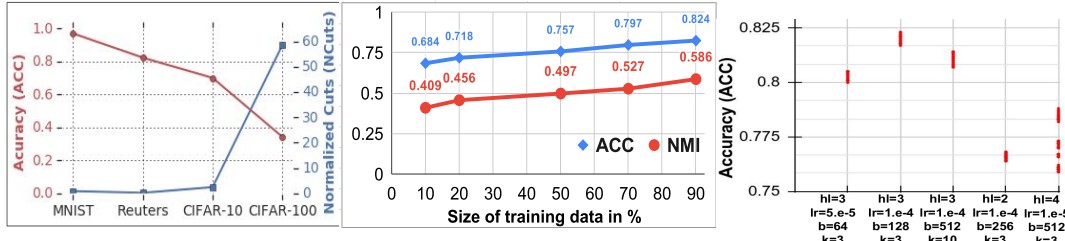

Figure 2: Normalized cuts (right axis) and clustering accuracy (left axis) are anti-correlated. Lower normalized cuts results in better accuracy.

Figure 3: Reuters: with only 10% training data the ACC and NMI of *CNC* are only 14% and 18% lower than ACC and NMI with 90% training data.

Figure 4: Reuters: CNC is trained by fixing some parameters and varying others. With $lr = 5e - 4, b = 128, k = 3$: ACC is $0.821 \pm 4e - 3$.

### 5.7 MODEL ARCHITECTURE AND HYPER-PARAMETERS:

Here are the details of the *CNC* model for each dataset.

- MNIST: The Siamese network has 4 layers sized [1024, 1024, 512, 10] with ReLU (Embedding size d is 10). The clustering module has 2 layers sized [512, 512] with a final gumbel softmax layer. Batch sized is 256 and we only consider 3 nearest neighbors to find the embeddings and constructing the affinity graph for each batch. We use Adam with lr = 0.005 with decay 0.5. Temperature starts at 1.5 and the minimum is set to 0.5.

- Reuters: The Siamese network has 3 layers sized [512, 256, 128] with ReLU (Embedding size d is 128). The clustering module has 3 layers sized [512, 512, 512] with tanh activation and a final gumbel softmax layer. Batch sized is 128 and we only consider 3 nearest neighbors to find the embeddings and constructing the affinity graph for each batch. We use Adam with lr = 1e-4 with decay 0.5. Temperature starts at 1.5 and the minimum is set to 1.0.

- CIFAR-10: The Siamese network has 2 layers sized [512, 256] with ReLU (Embedding size d is 256). The clustering module has 2 layers sized [512, 512] with tanh activation and a final gumbel softmax layer. Batch sized is 256 and we only consider 2 nearest neighbors to find the embeddings and constructing the affinity graph for each batch. We use Adam with lr = 1e-4 with decay 0.1. Temperature starts at 2.5 and the minimum is set to 0.5.

- CIFAR-100: The Siamese network has 2 layers sized [512, 256] with ReLU (Embedding size d is 256). The clustering module has 3 layers sized [512, 512, 512] with tanh activation and a final gumbel softmax layer. Batch sized is 1024 and we only consider 3 nearest neighbors to find the embeddings and constructing the affinity graph for each batch. We use Adam with lr = 1e-3 with decay 0.5. Temperature starts at 1.5 and the minimum is set to 1.0.

**Hyper-parameter Sensitivity:** We train the *CNC* on the Reuters dataset by fixing some hyper-parameters and varying others. We noticed that CNC benefits from tuning the number of hidden layers (hl), learning rate (lr), batch size (b), and the number of nearest neighbors (k), but is not particularly sensitive to any of the other hyper-parameters, including decay rate, patience parameter (cadence of epochs where decay is applied), Gumbel-Softmax temperature or minimum temperature (Figure 4). More precisely, we varied decay rate over the range [0.1-1.0], patience from [5-25] epochs, Gumbel-Softmax temperature from [1.0-2.0], and minimum temperature from [0.5-1.0]. When we fix hl=3, lr=5e-5, b=64, and k=3, the average accuracy is $0.803 \pm 2e - 3$. With hl=3, lr=5e-4, b=512, and k=10, the average accuracy is $0.811 \pm 2e - 3$. With hl=3, lr=5e-4, b=128, and k=3, the average accuracy is $0.821 \pm 4e - 3$. With hl=2, lr=1e-4, b=256, and k=3, the average accuracy is $0.766 \pm 9e - 4$. And finally with hl=4, lr=1e-5, b=512, and k=3, the average accuracy is $0.766 \pm 7e - 3$. As one can see, the accuracy varied from 0.766 to 0.821.

## 6 CONCLUSION

We propose *CNC* (*Clustering* by learning to optimize *Normalized Cuts*), a framework for learning to cluster unlabeled examples. We define a differentiable loss function equivalent to the expected normalized cuts and use it to train *CNC* model that directly outputs final cluster assignments. *CNC* achieves state-of-the-art results on popular unsupervised clustering benchmarks (MNIST, Reuters, CIFAR-10, and CIFAR-100 and outperforms the strongest baselines by up to 10.9%. *CNC* also enables generation, yielding up to 21.9% improvement over SpectralNet (Shaham et al., 2018), the previous best-performing generalizable clustering approach.

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

# 7 APPENDIX

Graph partitioning is a well-studied subject in computer science (Karypis & Kumar, 2000; Karypis et al., 1999; Karypis & Kumar, 1998; Miettinen et al., 2006; Sanders & Schulz, 2013; Andersen et al., 2006; Chung, 2007) with numerous applications in computer vision, VLSI design, biology, social networks, transportation networks and more.

In this section, we evaluate the performance of the *CNC* to partition computational graphs. In such graphs, nodes represent operations (e.g., MatMul, Conv2d, Sum) and edges represent the computational flow. Partitioning of the computational graph can be used for efficient mapping of the computation across the underlying hardware (e.g., CPUs and GPUs). Minimizing the normalized cuts translates to less communication between devices and balanced computation on each device.

## 7.1 DATASETS AND BASELINE METHODS

We conducted experiments on five widely used TensorFlow computation graphs: *ResNet*, *Inception-v3*, *AlexNet*, *MNIST-conv*, and *MNIST-conv*. Clustering of operations can be used for the mapping of the computation across the underlying hardware (e.g., CPUs and GPUs).We use the open source partitioners hMETIS (Karypis & Kumar, 2000) and KaHIP (Sanders & Schulz, 2013), a family of graph partitioning programs based on (Sanders & Schulz, 2012b;a) to find high quality ground truth partitions on TensorFlow computation graphs. More specifically, KaHIP includes KaFFPa (Karlsruhe Fast Flow Partitioner) and KaFFPaE (KaFFPaEvolutionary). KaFFPaE is a parallel evolutionary algorithm that uses KaFFPa's combine and mutation operations, as well as KaBaPE which extends the evolutionary algorithm. We compare the generalization results of *CNC* against the best results among the partitioners. The details of the datasets are as follows:

- *ResNet* (He et al., 2016) is a deep convolutional network with residual connections. The TensorFlow implementation of *ResNet_v1_50* with 50 layers contains 20,586 operations.

- *Inception-v3* (Szegedy et al., 2017) consists of multiple blocks, each composed of several convolutional and pooling layers. The TensorFlow graph of this model contains 27,114 operations.

- *AlexNet* (Krizhevsky et al., 2012) consists of 5 convolutional layers, some of which are followed by maxpool layers, and 3 dense layers with a final softmax. The TensorFlow graph of this model has 798 operations.

- *MNIST-conv* has 3 convolutional layers. Its TensorFlow graph contains 414 operations.

- *VGG* (Simonyan & Zisserman, 2014) has 16 convolutional layers. The TensorFlow graph of *VGG* contains 1,325 operations.

## 7.2 PERFORMANCE MEASURES

To evaluate the quality of *CNC*'s clusterings, we use two commonly used performance metrics in graph partitioning: 1) *Edge cut (Cut)*: the ratio of the cut to the total number of edges, and 2) *Balancedness (Bal)*: one minus the MSE between the number of nodes in every cluster and the number of nodes in an ideal balanced cluster ($\frac{n}{g}$). Both *Cut* and *Bal* are between 0 and 1. A lower *Cut* is better while a higher *Bal* is better. The *CNC* loss function in Equation 6 only considers optimizing the expected normalized cut. We add a regularizer to improve balancedness between clusters.

|              | AlexNet | | Inception-v3 | | ResNet | |
|--------------|-------|-------|-------|-------|-------|-------|
| Embedding    | Cut   | Bal   | Cut   | Bal   | Cut   | Bal   |
| None         | 0.166 | 0.717 | 0.242 | 0.740 | 0.450 | 0.908 |
| GCN          | 0.078 | 0.996 | 0.123 | 0.984 | 0.110 | 0.941 |
| GraphSAGE-off| 0.070 | 0.998 | 0.088 | 0.992 | 0.093 | 0.959 |
| GraphSAGE-on | 0.069 | 0.998 | 0.064 | 0.987 | 0.084 | 0.983 |

Table 4: Generalization results: *CNC* is trained on *VGG* and validated on *MNIST-conv*. During inference, the model is applied to unseen TensorFlow graphs: *ResNet*. *Inception-v3*, and *AlexNet*. The ground truth for *AlexNet* is *Bal* = 99%, *Cut* = 4.6%, for *Inception-v3*, is *Bal* = 99%, *Cut* = 3.7%, and for *ResNet* is *Bal* = 99% and *Cut* = 3.3%. *GraphSAGE-on* generalizes better than the other models.

## 7.3    EXPERIMENTAL RESULTS

To show that *CNC* generalizes effectively on unseen graphs, we train *CNC* on a single TensorFlow graph, *VGG*, and validate on *MNIST-conv*. During inference, we test the trained model on unseen TensorFlow graphs: *AlexNet*, *ResNet*, and *Inception-v3*. We consider the best quality result among hMETIS, KaFFPa, and KaFFPaE as the ground truth. The ground truth for *AlexNet* is *Bal* = 99%, *Cut* = 4.6%, for *Inception-v3*, is *Bal* = 99%, *Cut* = 3.7%, and for *ResNet* is *Bal* = 99% and *Cut* = 3.3%.

Table 4 shows the result of our experiments, and illustrates the importance of graph embeddings in generalization. The operation type (such as Add, Conv2d, and L2loss in TensorFlow) is used as the node feature as a one-hot. We leverage GCN (Kipf & Welling, 2017) and GraphSAGE (Hamilton et al., 2017) to capture similarities across graphs. In *GraphSAGE-on* both node embedding and clustering modules are trained jointly, while in *GCN* and *GraphSAGE-off*, only the clustering module is trained. Table 4 shows that the *GraphSAGE-on* (last row) achieves the best performance and generalizes better than the other models. Note that this model is trained on a single graph, *VGG* with only 1325 nodes, and is tested on *AlexNet*, *ResNet*, and *Inception-v3* with 798, 20586, and 27114 nodes respectively. On the other hand, the ground truth is the result of running different partitioning algorithms on each graph individually. In this work, our goal is not to beat the existing graph partitioning algorithms which involve a lot of heuristics on a given graph. Our generalization results show promise that rather than using heuristics, *CNC* is able to learn graph structure for generalizable graph partitioning.

*Model Architecture and Hyper-parameters:* The details of the model with the best performance (*GraphSAGE-on*) are as follows: the input feature dimension is 1518. GraphSAGE has 5 layers sized 512 with shared pooling, and the graph clustering module has 3 layers sized 64 with a final softmax layer. We use ReLU, Xavier initialization (Glorot & Bengio, 2010), and Adam with lr = 7.5e-5.

