# OpenReview forum: "Generalized Clustering by Learning to Optimize Expected Normalized Cuts"
_ICLR.cc/2020/Conference — Reject_

### Official Review · AnonReviewer2 · 2019-10-13
**Official Blind Review #2**

**Rating:** 6

**Review:**

This paper proposes a new clustering method, called CNC, which is composed of two-step procedures.
It first embeds an input dataset into a d-dimensional space, followed by performing relaxed normalized cut to detect clusters.
Although the contribution of introducing a new relaxed formulation of the normalized cut is interesting, I have the following concerns regarding with the clarity, significance, and evaluation of the proposed method.

- The paper is not clearly written at many points and the quality of presentation is not high, which also deteriorates the significance of the paper.
    In particular, the optimization process for clustering discussed in Section 4.1 is not clearly presented.
    Although the objection function, which is the expectation of the Ncut, is introduced in Equation (6), how to solve it is not presented.
    Since this is the key step for CNC, it should be carefully discussed.
- In the embedding step, how to choose the dimensionality d?
    This is not even reported in experiments.
- Empirical evaluation is not thorough and important evaluation is missing.
    * First, the contribution of embedding is not evaluated.
      The performance between CNC with the proposed embedding and without it should be compared.
      Moreover, the sensitivity of the performance with respect to changes in d should be examined.
    * A number of resulting scores are missing; in particular, CNC is compared to only SpectralNet for CIFAR-10 and CIFAR-100 under NMI.
      It would be more convincing if the NMI for other methods are also reported.
- Parameter sensitivity is not evaluated, while there are a number of parameters in the proposed method as reported in Section 5.7.
    Since parameter tuning is fundamentally difficult in the unsupervised setting, parameter sensitivity is crucial.
    Also how to choose such parameters is not clear.

Minor comments:
- In Algorithm 1, line 1, "X \in R^n" -> "X \subseteq R^m"?
- In Algorithm 1, the dimensionality "m" of data points and the batch size "m" are the same. Is it correct?
- At the first line in Section 4.1: "for each data point" -> "For each data point"
- P.4, L.-4: "nreast-neighbor" -> "nearest-neighbor"


**Experience Assessment:**

I have published one or two papers in this area.

**Review Assessment: Checking Correctness Of Derivations And Theory:**

I assessed the sensibility of the derivations and theory.

**Review Assessment: Checking Correctness Of Experiments:**

I carefully checked the experiments.

**Review Assessment: Thoroughness In Paper Reading:**

I read the paper at least twice and used my best judgement in assessing the paper.

---

> ### Author Response · Authors · 2019-11-14
> **Response to Reviewer #2**
>
> Thank you for your thorough and insightful feedback! We rewrote Section 4.1 to make it clearer, ran several additional experiments to address your concerns, and updated the paper accordingly. We have made significant effort to address all of your feedback and we believe that the paper is much stronger as a result, so we hope you will consider increasing your score.
>
> **Clarity and more details on the optimization process:
> We rewrote Section 4.1 to provide more details about the CNC model, as well as a clearer explanation of Algorithm 1. We also clarified that the CNC model F_{\theta} is implemented using a neural network, where the parameter \theta denotes the network weights (y_i = F_{\theta}(v_i)). CNC is trained to optimize the objective presented in Equation 7 via backpropagation.
>
> **Choosing the embedding dimension d:
> We updated Section 5.7 to clarify that dimension d is the dimension of the last layer of the Siamese network and specified the d values used for each of the datasets. Our novelty arises from the clustering phase in Algorithm 1 (steps 4-8). To allow for a fair comparison with SpectralNet (the prior state of the art method) and to evaluate the effectiveness of our clustering method, we used the same Siamese network architecture as SpectralNet on datasets on which they reported results.
>
> **Contribution of the embedding:
> We performed an ablation study to evaluate the impact of embeddings by omitting the embedding step in Algorithm 1. While our results confirm that embeddings contribute to better performance, CNC still achieves significant gains over SpectralNet  (Shaham et al., 2018)  when both are evaluated without embeddings. These results have been added to Section 5.3.
>          MNIST: The ACC (Accuracy), and NMI (Normalized Mutual information) are 0.945 and 0.873, while with the embedding, ACC and NMI increase to 0.972 and 0.924. Even without the embedding, our method outperforms SpectralNet (ACC is 0.8, and NMI is 0.814) reported in (Shaham et al., 2018).
>         Reuters: The ACC and NMI are 0.684 and 0.428, but with the embedding, ACC and NMI increase to 0.824 and 0.583. Even without the embedding, our results outperform those of SpectralNet (ACC is 0.605, and NMI is 0.401) reported in (Shaham et al., 2018).
>
> **Parameter sensitivity:
> We thank the reviewer for this insightful comment. We ran new experiments to measure the sensitivity of CNC to different hyper-parameters, leading to some interesting observations.
> We added analysis of these results, and a figure for visualization (Figure 4 in Section 5.7). To evaluate the sensitivity of the model to different hyper-parameters, we train the CNC model on the Reuters dataset by fixing some hyper-parameters and varying others. We noticed that CNC benefits from tuning the number of hidden layers (hl), learning rate (lr), batch size (b), and the number of nearest neighbors (k), but is not particularly sensitive to any of the other hyper-parameters, including decay rate, patience parameter (cadence of epochs where decay is applied), Gumbel-Softmax temperature or minimum temperature.
> More precisely, we varied decay rate over the range [0.1-1.0], patience from [5-25] epochs, Gumbel-Softmax temperature from [1.0-2.0], and minimum temperature from [0.5-1.0]. Here are the results when we fixed the learning rate (lr), batch size (b), and number of nearest neighbors (k):
> ++ hl=3, lr=5e-5, b=64, and k=3 : the average accuracy is 0.803 with standard deviation 2e-3.
> ++ hl=3, lr=5e-4, b=128, and k=3, the average accuracy is 0.821 with standard deviation 4e-3.
> ++ hl=3, lr=5e-4, b=512, and k=10, the average accuracy is 0.811 with standard deviation 2e-3.
> ++ hl=2, lr=1e-4, b=256, and k=3, the average accuracy is 0.766 with standard deviation 9e-4.
> ++ hl=4, lr=1e-5, b=512, and k=3, the average accuracy is 0.766 with standard deviation 7e-3.
> As you can see, the accuracy varied from 0.766 to 0.821, a difference of only 0.05.
>
> **Regarding missing NMI for some baselines in Table 1:
> Only recent clustering methods have reported NMI, and before that, accuracy (ACC) was the only metric for comparison. For thoroughness, we have reported both NMI and ACC for our method and included NMI for all methods that reported it.

---

### Official Review · AnonReviewer1 · 2019-10-21
**Official Blind Review #1**

**Rating:** 6

**Review:**

This paper presents an end-to-end approach for clustering. The proposed model is called CNC. It simultaneously learns a data embedding that preserve data affinity using Siamese networks, and clusters data in the embedding space. The model is trained by minimizing a differentiable loss function that is derived from normalized cuts. As such, the embedding phase renders the data point friendly to spectral clustering.
The paper follows the general setup of deep clustering: map data to a feature space while maintaining data distributional characteristic, and make data clustering-friendly in the feature space. The authors use Siamese networks for the first part and use a normalized-cut motivated loss for the second part.  The choices are reasonable and the loss is somewhat novel.
CNC is evaluated on standard datasets, including MNIST, Reuters, CIFAR-10, and CIFAR-100. The results are impressive. However, deep clustering has been around for quite a few years. It might be time to move on to more challenging benchmarks.


**Experience Assessment:**

I have published one or two papers in this area.

**Review Assessment: Checking Correctness Of Derivations And Theory:**

I assessed the sensibility of the derivations and theory.

**Review Assessment: Checking Correctness Of Experiments:**

I assessed the sensibility of the experiments.

**Review Assessment: Thoroughness In Paper Reading:**

I read the paper at least twice and used my best judgement in assessing the paper.

---

> ### Author Response · Authors · 2019-11-14
> **Response to Reviewer #1**
>
> Thank you for your very helpful suggestions! We added new results in the Appendix on additional benchmarks. Given that we have made significant effort to address your concerns and greatly improved the paper, we hope you will consider increasing your score.
>
> **Adding more challenging benchmarks for deep clustering beyond MNIST, Reuters, and CIFAR:
> To address your comment, we evaluated CNC on the task of partitioning computational graphs, comparing against state-of-the-art graph partitioning baselines. In clustering, the affinity graph is built implicitly over the data points, whereas in graph partitioning, an explicit graph is given by the adjacency matrix. Previous benchmarks (MNIST, Reuters, and CIFAR) required generalization to unseen data points (nodes), whereas with these new benchmarks, we explore whether the CNC model is able to generalize to unseen graphs. We believe this is a more challenging task, because the model must learn to capture the graph structure, rather than merely learning to represent individual nodes. Based on new experiments that we ran, the CNC model demonstrates promising results on generalizable graph partitioning. In order to better capture the graph structure, we had to improve our embedding step by using graph embeddings, such as GraphSage and GCN (our best results were with GraphSage). For further details, please see the full experimental results in the Appendix. By evaluating deep clustering methods against these more challenging benchmarks, as suggested by the reviewer, we hope to push the field of deep clustering forward and investigate avenues for additional impact.

---

### Official Review · AnonReviewer3 · 2019-10-23
**Official Blind Review #3**

**Rating:** 6

**Review:**

The paper suggests a differentiable objective that can be used to train a network to output cluster probabilities for a given datapoint, given a fixed number of clusters and embeddings of the data points to be clustered. In particular, this objective can be seen as a relaxation of the normalized cut objective, where indicator variables in the original formulation are replaced with their expectations under the trained model. The authors experiment with a number of clustering datasets where the number of cluster is known beforehand (and where, for evaluation purposes, the ground truth is known), and find that their method generally improves over the clustering performance of SpectralNet (Shaham et al., 2018) in terms of accuracy and normalized mutual information, and that it finds solutions with lower normalized cut values.

The method proposed in this paper is very simple and appears to work well, and so this paper represents an important contribution. However, there are some issues with the presentation that I think should be fixed before publication:
- Equation (3): I'm not sure I understand the sum over z; don't we just want w_{ij} Y_{ik} (1 - Y_{jk})?
- Equation (6): I don't think the final objective should be presented as an expectation. It is rather the quotient of two expectations. In general, it might be better to just present the objective as a relaxation of the normalized cut objective.

A question regarding the results and parameterization: was using Gumbel-Softmax necessary to get good results? Did ordinary softmax not work?

**Experience Assessment:**

I have read many papers in this area.

**Review Assessment: Checking Correctness Of Derivations And Theory:**

I assessed the sensibility of the derivations and theory.

**Review Assessment: Checking Correctness Of Experiments:**

I assessed the sensibility of the experiments.

**Review Assessment: Thoroughness In Paper Reading:**

I read the paper at least twice and used my best judgement in assessing the paper.

---

> ### Author Response · Authors · 2019-11-14
> **Response to Reviewer #3**
>
> Thank you so much for your valuable suggestions which helped us improve the paper!  We believe that we have addressed all of your concerns, so we hope you will consider increasing your score.
>
> **Clarifying Equations 3, 6:
> We updated Equations 3, 5, and 6 in Section 4.1. We agree with the reviewer about the Equation 3 (thanks for pointing this out!). In the attached implementation, the formula was correct and we have now updated the paper to reflect the comment. To better show how volume affects the expected normalized cuts, we added Equation 5 as an intermediate step to the final loss in Equation 6. We also added Equation 7 and we clarified that the CNC model F_{\theta} is implemented using a neural network, where the parameter \theta denotes the network weights (y_i = F_{\theta}(v_i)). CNC is trained to approximate Equation 7 via backpropagation.
>
> **Use of Gumbel-Softmax:
> In our experiments, we first tried regular Softmax, but found that the model quickly converged to a local minimum, becoming overconfident in sub-optimal assignments of nodes to clusters. However, the high temperature in the early stage of training with Gumbel-Softmax allows the model to explore different cluster assignments and gradually become more confident as we decrease the temperature.

---

### Author Response · Authors · 2019-11-14
**Response to all Reviewers**

We would like to thank the reviewers for their valuable feedback which we have used to greatly improve the paper!

We are glad that the reviewers felt that “this paper represents an important contribution” and that “the results are impressive”, and we have made significant effort to address all of the reviewers’ concerns, including running new experiments, rewriting sections of the paper, and open-sourcing our framework for reproducibility.

For clarity, we have also highlighted the revised sections in this document (https://www.dropbox.com/s/fd0ujw5n6898ldo/CNC_ICLR20_revision.pdf?dl=0)

++ Regarding clarity and explanation of the optimization process, we have updated Section 4.1:
    1) We further describe the CNC model architecture and provide a more detailed explanation of Algorithm 1.
    2) We added intermediate steps in Equations 5 and 6 to illustrate the derivation of our final loss function.
    3) We added further explanation of how CNC optimizes for our objective function via backpropagation.

++ Regarding the contribution of the embedding step, we have updated Section 5.3:
    1) We performed an ablation study to evaluate the impact of the embeddings by omitting the embedding step.

++ Regarding hyper-parameter sensitivity, we have updated Section 5.7:
    1) We clarified that the last layer of the Siamese network represents embedding size d.
    2) We added analysis of the hyper-parameter sensitivity of the CNC model.

++ Regarding more challenging benchmarks for deep clustering beyond MNIST, Reuters, and CIFAR, we added an Appendix:
    1) We have evaluated the performance of CNC on computational graphs and our results indicate that CNC generalizes
         effectively to unseen graphs. The experimental results have been added to the Appendix.

---

### Decision · Program_Chairs · 2019-12-19

**Decision:**

Reject

**Comment:**

This paper proposes a deep clustering method based on normalized cuts.  As the general idea of deep clustering has been investigated a fair bit, the reviewers suggest a more thorough empirical validation.  Myself, I would also like further justification of many of the choices within the algorithm, the effect of changing the architecture.